

# Stable isotope investigation of groundwater recharge in the Carpathian Mountains, East-Central Europe

Carmen - Andreea Bădăluță[1,2,3,4], Aurel Perşoiu [1,5,6], Monica Ionita[4], Viorica Nagavciuc[1,3,7,8], Petruţ - Ionel Bistricean[2,9]

[1]Stable Isotope Laboratory, Ștefan cel Mare University of Suceava, Suceava , Romania

[2]Department of Geography, Stefan cel Mare University of Suceava, Suceava, Romania

[3]Institute for Geological and Geochemical Research, Research Centre for Astronomy and Earth Sciences MTA, Budapest, Hungary

[4]Alfred Wegener Institute, Helmholtz Center for Polar and Marine Research, Bremerhaven, Germany

[5]Emil Racoviţă Institute of Speleology, Cluj Napoca, Romania

[6]Institute of Biology, Department of Microbiology, Bucharest, Romania

[7]Faculty of Forestry, Stefan cel Mare University of Suceava, Suceava, Romania

[8]Departement of Geography, Johannes Gutenberg University, Mainz, Germany

[9]Regional Meteorological Center of Moldova, Suceava, Romania

*Correspondence to*: C.-A. Bădăluţă (carmenbadaluta@yahoo.com) and A. Perşoiu (aurel.persoiu@gmail.com)

## Abstract

Rapid growth in water usage in NW Romania has led to an increased pressure on the available water resources; however, the relationships between precipitation, surface and groundwater in the region are poorly understood. Here, we have analyzed the stable isotopes of oxygen and hydrogen in precipitation, river and groundwater to gain information on moisture sources feeding precipitation in the area and establish the main links between the large-scale atmospheric circulation, precipitation amount and discharge. Thus, in this study we have analyzed 157 groundwater samples, 64 precipitation samples from two collection sites (one in mountain area and another one in plateau area) and 54 rivers samples from two rivers. Furthermore, we have directly linked the changes in the isotopic composition of the d-excess parameter in the precipitation with the processes linked to large-scale atmospheric circulation. Isotopes in precipitation water resulted in two



LMWLs ($\delta^2$H=7.4*$\delta^{18}$O+2.7 at 350 m asl and $\delta^2$H=8.1*$\delta^{18}$O+12.4 at 1530 m asl), with a clear seasonal signal, further enhanced by secondary evaporative processes in summer. Moisture in the lowlands was mostly delivered along easterly trajectories, while that in the mountain area from the westerlies. Surface water analyses show the same trend as precipitation, but with reduced amplitude between summer and winter

values. Throughout the winter season, the $\delta_{prec}$ is strongly related with different climate teleconnection patterns like the East Atlantic (EA), the North Atlantic Oscillation (NAO) and the Arctic Oscillation (AO), while during summer, the $\delta_{prec}$ shows a strong correlation with the Atlantic Multidecadal Oscillation (AMO) and the summer EA. Maps of $\delta^{18}$O and d-excess distribution in groundwaters show a depletive trend from NW to SE, generated in principal by topography. The waters in the aquifers show no clear patterns and

altitude effect.

## 1. Introduction

Water availability is a major concern amid growing population, intensified agriculture and industrial growth

(Vörösmarty et al., 2010). The competition between these, on the backdrop of ongoing climate changes, exerts an increasing pressure on available water sources, generally leading to conflicts between users, depletion of resources and social and economic losses (Jury and Vaux, 2007). Numerous studies have addressed these issues in different parts of the globe; however, none such was conducted in the wider Carpathian region (Romania, Hungary, Ukraine, Moldavia), home to more than 80 million people and one of

the most important agricultural regions of Europe. The region is drained by the Danube River, the second largest European river (after Volga), its tributaries in the area being fed by moisture thought to come from both the North Atlantic Ocean and the Mediterranean sea (Ciric et al., 2016). Suceava River, located in NE Romania (East-Central Europe), is one of the major tributaries of Siret and the largest Carpathian tributary of the Danube River. Surface and groundwater from the river's watershed is the main water source for more

than 1 million people and agriculture in Suceava, Botoșani and Iași counties (fig. 1). Iași, the second largest urban area in Romania after Bucharest (Eurostat), and one of the few cities in the country that has positive demographic balance, partly relies on groundwater from the Suceava watershed for its domestic consumption. More than half of Suceava county's population (634,000 inhabitants) also uses surface and ground water from the same watershed for domestic purposes. Further, Suceava and the neighbouring

Botoșani counties are hosting the largest bovine populations in Romania, and Suceava is home to the largest



forested area in Romania (and associated lumber industry). Local populations are competing with agriculture, natural vegetation and industry, exerting a continuously increasing pressure on the water resources. This problem is exacerbated by the increasing use of groundwater for irrigation in Romania, the country being the 5[th] exporter of groundwater per capita in the world (Dalin et al., 2017). The continental

climate of the region (with hot and dry summers, and an eastward increasing soil moisture deficit) adds powerful constrains on the agricultural use of the land, sometimes with catastrophic crop failures. Further, over the past decades, the studied region experienced a fast and strong winter warming, average winter temperatures increasing by 2.5 °C between 1961 and 2007 (Busuioc et al., 2012, figure 1b). As such, irrigations are strongly needed to maintain productivity, while forecasted droughts will act to limit it.

Nevertheless, the disintegration of the communist block in 1989 and ascension in the EU in 2007 led to political and economical possibilities and incentives (relocation of agriculture from high-income to low-income countries, subsidies, compaction of scattered plots and farms, improved techniques and crops) for the development of agriculture in the region. However, while policy and economy are changing, so does climate: warming and intensification of extremes are unequivocal (Collins et al., 2013) over the past decades, and

climate models suggest that these trends are likely to shape a changed climate in the future (Collins et al., 2013). Recent studies (Busuioc et al., 2014; Rîmbu et al., 2014; Bîrsan et al., 2014; Cheval et al., 2014; Ionita et al., 2016) have convincingly shown that the region is experienced a continuous warming and drying trend in the past decades, both during winter and summer, that have also affected rivers' discharge (Ionita et al. 2015a; Ionita, 2015). Changing climate can affect the water balance of river catchments, via changes in

precipitation (which affect the river runoff) and via increased temperatures which lead to increased evapotranspiration, as well as melting of the glaciers and changes in the timing and amount of snow fall and snow melt. As a consequence, the summer runoff is expected to decrease over the southern part of Europe (including Romania) (Leipprand et al., 2008). Combining the above facts, it results that the intensity of agricultural use of the land and population increase will led to a surge in water usage under warming and

drying climatic conditions, resulting in increased stress on already limited water resources. Given the above, a clear understanding of the moisture sources for surface and ground waters as well as their relation with large-scale circulation patterns is required if the competing interest in water usage are to be addressed to the benefit of all users (e.g., Wassenaar et al., 2011).

In hydrology, the stable isotopes of hydrogen and oxygen are established tools in tracing sources and fluxes

of water from source to precipitation and surface/ground water (Bowen, 2010; Clark and Fritz, 1997;



Ferguson et al., 2007). The links between the stable isotope distribution of O and H in water and climate are well understood in general principles and these can be locally applied to distinguish between different moisture sources and tracks, seasonal contribution to river and groundwater recharge (ref), post-precipitation processes (e.g., evaporation) etc. However, so far, no such studies have been performed in our study area,

5    and, as a matter of fact, in Romania, except for a few studies aimed at understanding the stable isotope composition of precipitation in Western Romania (Bojar et al., 2009; Bojar et la. 2017; Drăguşin et al., 2017). As such, the objectives of our study are to identify, by analyzing the stable isotope composition of precipitation, surface and groundwater, the principal moisture sources feeding rainfalls in NW Romania and understand their role in recharging local aquifers.

**2. Study area**

Suceava river (170 km long) has its headwaters at 1508 m above sea level (asl), in Obcina Mestecăniş Mountains, and about ¼ of the drainage basin (3800 km$^2$) extends in mid-altitude mountains, with the remaining one draining a low-altitude (300-500 m asl) tableland (Figure 1). The mean annual discharge is

16.1 m$^3$s$^{-1}$, varying between 0.94 m$^3$s$^{-1}$ and 1700 m$^3$s$^{-1}$, the minimum values occurring in early autumn, and the maximum in early summer, driven by the precipitation maximum that occurs in June. Soloneţ (31 km long) is the second tributaries to right of Suceava (210.7 km$^2$) and is located in Suceava Plateau (Bădăluţă et al., 2013). The mean annual discharge is 1.3 m$^3$s$^{-1}$ and varies between 0.1 m$^3$s$^{-1}$ and 137 m$^3$s$^{-1}$, with a regime similar to that of Suceava River.

The climate in Suceava River's watershed is temperate-continental, the mean annual temperature being around 8 °C, with a minimum in January and maximum in July, slightly lower in the western, mountainous sector (Brânduş and Cristea, 2013). The mean annual precipitation is 606.6 mm, being slightly higher in west – 700 mm at Izvoarele Sucevei, and lower in east – 580 mm at Liteni  (Cocerhan, 2012); with 73.5 % of annual precipitation falling between April and September.

**3. Data and Methods**

Precipitation samples were collected monthly between December 2012 and December 2016 at Rarău (RR, 47°27'N, 25°34'E, 1536 m asl) and Suceava (SV, 47°38'N, 26°14'E, 352m asl) stations (Figure 1), both within the catchment of Suceava River. River water was collected monthly between November 2014 and

December 2016, from Suceava River at Suceava (Figure 1) and from its tributary, Soloneţ, ca. 20 km west of



Suceava (Figure 1). A total of 157 groundwater samples were collected seasonally in October 2014, February 2015, May 2015 and August 2015 from 21 dug wells (between 2 and 18 m deep) spread throughout the Suceava River Basin (Figure 1). Well and river samples were collected as grab samples (30-50 cm below the surface of the water) in 20 ml HDPE scintillation vials. Precipitation water was collected continuously using

self-built collectors, constructed according to IAEA specification. A 5 liter HDPE plastic canister is fitted with a funnel, prolonged with a plastic tube, channeling water to the bottom of the container. Excess air escapes the canister through to a narrow, 3 m long plastic tube, to minimize air exchange between the container and the outside environment. Paraffin oil in the canister is used to minimize evaporation. The funnel is "sealed" with table tennis balls, to restrain insect access, but allow water collection. At the end of

each month a sample was taken from the container and stored in 20 ml HDPE scintillation vials at 4 °C until analysis. In winter, snow samples were collected into an open plastic container (10 l, 40 cm deep), allowed to melt at room temperature at the end of the month and stored as described above.

The water was analyzed for its stable isotope composition in the Stable Isotope Laboratory at the Ştefan cel Mare University (Suceava) using a Picarro L2130*i* CRDS analyzer coupled to a High Precision Vaporizing

module. Prior to analyses, all samples were filtered through 0.45 µm nylon membranes. Each samples was manually injected into to the vaporization module multiple times, until the standard deviation of the last four injections was less than 0.03 for $\delta^{18}O$ and 0.3 for $\delta^2H$. The average of these last four injections was normalized on the SMOW-SLAP scale using two internal standards calibrated against the VSMOW2 and SLAP2 standards provided by the IAEA. A third standard was used to check the long-term stability of the

analyzer. The stable isotope composition of oxygen and hydrogen are reported in the standard δ notation.

Daily climate (precipitation amount and air temperature) and discharge data were provided by the National Meteorological Administration and the Siret - Bacău Basin Waters Administration, respectively. The Hybrid Single Particle Lagrangian Integrated Trajectory (HYSPLIT) model (Stein et al., 2015) was used to reconstruct the trajectories of precipitation events larger than 3 mm (contributing >90 % of the total volume

of monthly precipitation). The model used the data generated by the Global Data Assimilation System (GDAS) and was set to compute trajectories backwards for 72 hours, at 500 m above ground level. This altitude was chosen as it is above local topographic influences and it is also close to the mean value of the cloud base during precipitation events.




For the large-scale atmospheric circulation we used the monthly means of Geopotential Height at 500mb (Z500), the zonal wind (U500) and the meridional wind (V500) at 500mb level from the Twentieth Century Reanalysis (V2) data set (NCEPv2, Whitaker et al. 2004; Compo et al. 2006, 2011) on a 2° × 2° grid, for the period 1871–2012.

The spatial variability of stable isotopes in groundwater was spatially mapped using the ordinary kriging method (Nas and Berktay, 2010).

To identify the physical mechanisms responsible for the connection between the winter and summer temperature at Suceava meteorological station and the large-scale atmospheric circulation, we constructed the composite maps of Z500 standardized anomalies for each season by selecting the years when the value of

10 the normalized time series of the winter and summer temperature was >1 Standard Deviation (High) and <−1 Standard Deviation (Low), respectively. This threshold was chosen as a compromise between the strength of the climate anomalies associated with the temperature anomalies and the number of maps that satisfy this criterion. Further analysis has shown that the results are not sensitive to the exact threshold value used for the composite analysis (not shown).

## 4. Results - stable isotopes in precipitation, rivers and groundwaters

The data set of monthly $\delta^{18}O$ and $\delta^2H$ in precipitation at the two investigated stations is shown in Table 1 and plotted in Figure 2, against air temperature and precipitation amount. At Suceava (SV, 352 m asl), the stable isotope composition of precipitation ($\delta_{prec}$) varies between 1.1 ‰ for $\delta^{18}O$ and -10 ‰ for $\delta^2H$ in summer; and

20 between -27.1 ‰ for $\delta^{18}O$ and -205 ‰ for $\delta^2H$ in winter, with mean values of -10.8 ‰ for $\delta^{18}O$ and -77 ‰ for $\delta^2H$. At Rarău (RR, 1536 m asl) the δ values are much lower, reflecting the altitude difference, being between -4.3 ‰ for $\delta^{18}O$ and -33 ‰ for $\delta^2H$ in summer, and between -21.7 ‰ for $\delta^{18}O$ and -163 ‰ for $\delta^2H$ in winter, with mean values of -11.8 ‰ for $\delta^{18}O$ and -84 ‰ for $\delta^2H$. The maximum deuterium-excess (d-excess) values (Table 1) are between 16.9 ‰ (December 2013, SV) and 22.2 ‰ (November 2013, RR),

while the minima are -3.6 ‰ (August 2013, RR) and -21 ‰ (February 2016, SV). The stable isotope composition of precipitation shows a good correlation with air temperature, with maxima in June-August and minima in December-February (Figure 2).

The Local Meteoric Water Lines (Figure 3) for the two stations are close to the Global Meteoric Water Line (GMWL, defined by the equation $\delta^2H=8.17*\delta^{18}O+10.35$; Craig, 1961, Rozanski et al., 1992), showing a





stronger influence of evaporation at Suceava ($\delta^2H=7.4*\delta^{18}O+2.7$), compared with the higher and wetter Rarău station ($\delta^2H=8.1*\delta^{18}O+12.4$).

Monthly river $\delta^{18}O$ and $\delta^2H$ ($\delta_{river}$) values are shown in Table 2 and plotted in Figure 4. The stable isotope composition of Suceava river water varied little between -8.8 ‰ and -10.8 ‰ for $\delta^{18}O$, and between -63 and

5  -75 ‰ for $\delta^2H$ (in August and February 2015, respectively); while in Soloneț river, δ values ranged between -8.2 ‰ and -10.7 ‰ for $\delta^{18}O$, and between -61 and -75 ‰ for $\delta^2H$ (in July 2016 and May 2015, respectively). The average values of $\delta^{18}O$ and $\delta^2H$ were -9.6 ‰ and -67 ‰ for both Suceava, and Soloneț, compared to -10.8 ‰ and -77 ‰ in precipitation at Suceava. The regression line between the plotted δ values gives a River Water Line (RWL) defined by the equations $\delta^2H=6.2*\delta^{18}O-7.2$ for Suceava and $\delta^2H=5.8*\delta^{18}O-$

11.4 for Soloneț, respectively (Figure 3). Low flow condition during July - September were characterized by maximum δ values, while during high flow conditions in spring and early summer, the rivers' water had average δ values. The d-excess values in river waters ranged from 5.1 ‰ to 12.3 ‰ (with minimum in summer), with a mean of 9.9 ‰ for Suceava River and 9 % for Soloneț River; both the extremes and means being lower than in precipitation (Table 1).

The stable isotope composition of groundwater ($\delta_{ground}$, Table 2) shows muted variability (between -8.5 and -10.6 ‰ for $\delta^{18}O$, and between -61 and -74 ‰ for $\delta^2H$) compared to precipitation and river water. Groundwater $\delta^{18}O$ and $\delta^2H$ values are rather stable throughout the year at most stations, with the exception of Cacica, Poieni Solca and Pârteștii de Jos. Groundwater d-excess values are also less variable than in precipitation in rivers, ranging from 6.6 ‰ to 11.5 % (with minimum in February and maximum in August)

and a mean of 9.8 ‰.

## 5. Discussion

### 5.1 Links between the stable isotope composition of precipitation water and regional air temperature, large-scale atmospheric circulation and moisture sources

The ~1100 m elevation difference between Suceava and Rarău stations translates into a very limited difference in the $\delta^{18}O$ and $\delta^2H$ values of precipitation: ~1 ‰ for $\delta^{18}O$ and ~7 ‰ for $\delta^2H$, suggesting an average gradient of 0.1 ‰/100 for $\delta^{18}O$, less than the average worldwide gradient of 0.26 ‰/100 calculated by Poage and Chamberlain (2001), and in partial agreement with values of 0.21 ‰/100 m in the nearby Carpathian Mts. of Slovakia (Holko et al., 2012). This difference is possibly due to local climatic conditions



in Suceava, the region having the coldest MAT in Romania for similar altitudes, as a result of strong Scandinavian and Western Siberian influences which results in lower than expected stable isotope values in precipitation for Suceava.

Analyses of the stable isotope-temperature relationship show a high correlation at both Suceava (r=0.73) and

Rarău (r=0.81), supporting this interpretation. The correlation between $\delta_{prec}$ and precipitation amount is lower (r=0.41 at Suceava and r=0.34 at Rarău). However, during summer, $\delta_{prec}$ and precipitation amount are positively correlated at Suceava (r=0.47) only, and not at Rarău (r=-0.06), suggesting that amount effect during heavy summer convective rains plays an important role in determining $\delta_{prec}$.

Air temperature variability in the study region is controlled by a complex interplay between the various

large-scale modes of climate variability controlling moisture delivery to the precipitation site. Analyses of correlation between air temperature and the main teleconnection indexes show a positive relationship with North Atlantic Oscillation (NAO), the East Atlantic (EA), East Atlantic/Western Russia (EA/WR), Atlantic Multidecadal Oscillation (AMO) and Artic Oscillation (AO) and negative with Polar/Eurasia (POL) and Scandinavia (SCA) indexes (Table 3). In winter, climatic conditions in Central and Eastern Romania (CEE)

are influenced by the NAO, the main mode of climatic variability in the Northern Hemisphere (Hurrel et al., 2013, Ionita et al., 2014), which in turn is influenced by the EA pattern that has an important role in the location and strength of NAO dipole (Moore et al. 2013). The positive phase of NAO (when the atmospheric pressure is below average in Iceland and above it in the Azores) is associated whit higher than normal temperatures in CEE and Southern Europe and the precipitation source is predominantly Atlantic.

Conversely, the negative phase of the NAO is linked to low temperatures in CEE and a southward displacement of the westerlies, carrying moisture from North Atlantic towards CEE and the Mediterranean Sea. This can be also observed when looking at the composite maps between the winter temperature at Suceava station (located in the north-eastern part of Romania) and the winter geopotential height at 500mb. Positive temperature anomalies are associated with a dipole like structure in the Z500 field, that projects onto

the positive phase of NAO (Figure 5a), but a little bit shifted towards Europe. This dipole-like structure, characterized by positive Z500 anomalies over the central North Atlantic Ocean extending up to the eastern Europe and negative Z500 anomalies centered around Iceland, favors the advection of warm and moist air over the analyzed region. Negative temperature anomalies in the north-eastern part of Romania, are associated with a center of positive Z500 anomalies around Iceland and a center of negative Z500 anomalies

over the whole central North Atlantic Ocean extending until the eastern part of Europe. This diploe-like



structure, associated with negative temperature anomalies over the analyzed region, projects well onto the negative phase of NAO (Figure 5b).

In summer, the action of atmospheric pressure centers on temperature and moisture sources is more complex than in winter, due to blocking structures and highly dynamic Rossby waves meandering over Europe (Ionita

et al., 2015, 2017, Schubert et al., 2014)). High temperatures are associated with a stationary anticyclonal structure over CEE in which the Rossby waves act in the convergence areas and the moisture sources is predominantly from Eastern Europe (Schubert et al., 2014). These modes of climate variability are affecting the $\delta_{prec}$ differently, with the EA, NAO and AO pattern having a stronger influence during winter (Figure 6), while during summer, $\delta_{prec}$ shows a strong correlation with the AMO and EA. Positive temperature

anomalies, over the north-eastern part of Romania, are associated with a classic "omega" blocking pattern (Dole and Gordon, 1983) characterized by a high pressure system over the whole eastern part of Europe (Figure 5d) and flanked by low pressure system on the left and right. In general, these kind of blocking situations are associated with heatwaves and droughts over the eastern part of Europe, like the exceptionally dry and warm summer of 2010 (Dole et al., 2011). Positive temperature anomalies tend to occur near the

center of the block, where northward displaced subtropical air, descending air motions and reduced cloudiness contribute to abnormally warm surface temperatures (Figure 5c). Negative summer temperature anomalies, over the north-eastern part of Romania, are associated with a wave-train in the Z500 field, with positive Z500 anomalies over the central North Atlantic Ocean and the British Isles and negative Z500 anomalies over the eastern part of Europe. This kind of pattern favors the advection of cold air from the

north towards the southern and eastern part of Europe (Figure 5d).

These differences are clearly discernible when identifying the moisture sources based on the analysis of deuterium excess (d-excess) values of precipitation water. D-excess values in precipitation are indicating changes in conditions at the moisture sources (or changes of the moisture sources), recycling processes along the moisture tracks (Jouzel and Merlivat, 1984) or reorganizations of the atmospheric circulation (Steffensen

et al., 2008). Analyses of the mean d-excess value show small differences between Rarău (11 ‰) and Suceava (10.3 ‰) stations, close to the average global value of 10 ‰. The maximum values of d-excess (between 12 and 18 ‰) are recorded in autumn at both stations, and the minimum ones (between 2 and 8 ‰) in winter (January-February) and spring. For better interpretation of d-excess values we analyzed the trajectories of the air masses delivering moisture to the study site, using the HYSPLIT back-trajectory model

(Stein et al., 2015). Between 2013-2016, 166 individual trajectories were computed (Figure 7 and Table 1).





Our data show that during spring and autumn most of the moisture at our site is coming along eastern trajectories or is locally recycled (42.7 % and 46.6 % of the total moisture delivered between 2013 and 2016, respectively), with the Atlantic Ocean contributing moisture mainly during spring (26.7 %) and summer (33.1 %). The Mediterranean and Black Seas are less important as moisture sources, the highest percentages occurring during winter and spring, when mobile cyclones are penetrating farther north. The relatively low percentages of Atlantic Ocean and Mediterranean Seas as moisture contributors are due to the orographic barrier effect of the Carpathians, with two mountains chains (Figure 1) blocking both Atlantic storm tracks and Mediterranean cyclones. Comparatively, stable isotopes studies in SW Romania (Bojar et al., 2009, Dragușin et al., 2017) and Hungary (Bottyán et al., 2017) have shown the strong influence of both Atlantic and Mediterranean sources for regions located in front of the mountains. High d-excess values in precipitation during autumns (12.2 ‰) are a further indicator of recycled moisture and/or a high-evaporative source, thus reinforcing Eastern Europe as the main source of precipitation east of the Carpathians Mountains chain (Vystavna et al., 2017). In summer and spring, the still important eastern component of precipitation is counterbalanced by the higher input from the Atlantic, and thus the high d-excess values in precipitation derived from the former are counterbalanced by the low values in Atlantic moisture (Table 3).

### 5.2. Stable isotopes in surface and groundwaters

The variability of $\delta_{river}$ follows a trend similar to that in precipitation, although strongly muted (Figure 4), with the amplitude of $\delta_{prec}$ being one order of magnitude larger than that of $\delta_{river}$ (24 ‰, compared to 2 ‰). The annual variability of $\delta_{river}$ in both rivers (Suceava and Soloneț) follows that in precipitation, with the minima in rivers occurring either at the same time as in precipitation or slightly delayed; the later case being specific for cold winters, when most of the precipitation is stored as snow. The maximum $\delta_{river}$ values occur in summer, but contrary to winter, they show a better correlation with air temperature, rather than with $\delta_{prec}$ (Figure 4), possibly indicating that the original stable isotope composition of river water was changed by evaporation. Evidence for this process comes from the d-excess values of river water, which show a strong positive correlation with discharge, with the lowest d-excess values occurring in summer, during minimum discharge. As expected, the d-excess of Soloneț, with a discharge one order of magnitude lower than that of Suceava, shows lower d-excess values, especially in summer, indicating the higher susceptibility to evaporation of the former. Discharge of Soloneț varied by more than 100 % between maximum and





minimum, the lowest values (summer 2016) being also the ones with the lowest d-excess (fig. xx, Ferguson et al., 2007, Wasesnaar et al, 2011). These processes are clearly shown by the alignment in the $\delta^{18}O$-$\delta^{2}H$ diagram of the river stable isotope values along two lines with slopes lower than the LMWL (figure 3). Non-equilibrium evaporation in low humidity condition during summer is responsible for the enrichment of the

surface waters in the heavy $^{18}O$ and $^{2}H$ isotopes (Froehlich et al., 2005) and the alignment of the samples along Local Evaporation Lines (LEL, Gibson et al., 2005), with their origin at the initial stable isotope composition prior to evaporation  (Rozanski et al., 2001). Higher rates of evaporation for the low-discharge Soloneț, compared to Suceava, resulted in a higher degree of enrichment in the heavy isotopologues for the former, as well as higher fractionation factors for both O and H (e.g., Gat, 208), resulting in lower slopes for

the LEL.

The $\delta^{18}O$ and $\delta^{2}H$ of groundwaters show similar values, with very limited variability, close to those in precipitation at the lower altitude Suceava station. No clear relationship with altitude and or position (latitude/longitude) has been found, and very limited seasonal differences were discerned, with samples collected in summer being the most depleted in heavy isotopes (although only by max. 0.5 ‰ for $\delta^{18}O$ and

1.5 ‰ for $\delta^{2}H$). The ground water d-excess values were more stable (between 8 and 12 ‰) than in precipitation and surface waters, again being slightly higher in samples collected in summer. All these point towards recharge from local precipitation of the groundwater and limited exchange with the rivers, as well as of a possible ca. 6 months residence time of waters, as indicated by the minimum values occurring in summer. Mapping of the spatial distribution of ground water $\delta^{18}O$ and d-excess values (figure 8) reveal no

clear pattern, except for a slight NW-SE decreasing gradient for $\delta^{18}O$, most evident in the warm months, which we tentatively link to 1) delayed recharge by snowmelt (see also above) and/or 2) slow underground flow on the same direction of the aquifer.

## 6. Conclusions

Stable isotopes in precipitation, surface waters and groundwater have been used to identify the moisture sources, transport mechanisms and hydrological processes responsible for groundwater recharge in NE Romania, a region of high-pressure on existing water resources. The HYSPLIT-modelled back trajectories combined with d-excess values of precipitation indicate that the main precipitation sources are located eastwards from the sampling site (in the East-European Plain and the Black Sea). At multiannual timescale,



the continental sources and locally recycled moisture contributes with 36% of the total rainfall, the share being higher in spring and autumn and lower in winter and summer. The LMWLs at the two precipitation collection stations are close to the GMWL, but with visible differences between the plateau ($\delta^2H=7.4*\delta^{18}O+2.7$) and mountainous ($\delta^2H=8.1*\delta^{18}O+12.4$) area, the former being strongly influenced by

5  intense evaporation processes, and the former by wetter (higher relative humidity), respectively. Secondary evaporation and moisture recycling further contributed to seasonality in precipitation. The $\delta^{18}O$ $\delta^2H$ values in surface waters were offset to higher values than those in precipitation. Evaporation strongly affects both investigated rivers, but to a higher degree for the smaller river and during periods of low flow for both Suceava and Soloneț rivers. Both surface and ground waters show a strong influence of winter precipitation

10  contribution to river discharge and aquifer recharge. Modelling data (Busuioc et al., 2012) indicate for the next two decades a 40 % decrease in winter and spring precipitation and possible increase in summer one, thus leading to a higher degree of continentally and a possible reduction in water availability, both at the surface and in the ground.





*Author contribution.* C-AB and VN designed the study, collected the samples and run the analyses, P-IB provided climate and hydrological data. C-AB and AP wrote the text with contribution from MI. All authors provided input on the final version of the text.

5   *Competing interests.* The authors declare that they have no competing interests.

*Acknowledgments.* This project was funded by grants number PN-II-RU-TE-2011-3-0235 and PN-II-RU-TE-2014-4-1993, financed by CNCS Romania and RO-18452, financed by the IAEA, awarded to AP. M.I. is funded by the Helmholtz Climate Initiative REKLIM.



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



*Table 1*. Monthly δ and d-excess values in pp and river

| Month | RR-O | RR-H | RR-d | SV-O | SV-H | SV-d | SVR-O | SVR-H | SVR-d | SVR-Q | SOL-O | SOL-H | SOL-d | SOL-Q |
|---|---|---|---|---|---|---|---|---|---|---|---|---|---|---|
| Dec-12 | | | | -19.1 | -137 | 15.9 | | | | | | | | |
| Jan-13 | | | | -18.1 | -133 | 11.4 | | | | | | | | |
| Feb-13 | | | | -13.8 | -99 | 11.6 | | | | | | | | |
| Mar-13 | | | | -15.6 | -112 | 13.6 | | | | | | | | |
| Apr-13 | | | | -7.2 | -47 | 10.3 | | | | | | | | |
| May-13 | -10.3 | -69 | 13.0 | -6.1 | -35 | 13.2 | | | | | | | | |
| Jun-13 | -10.6 | -73 | 11.8 | -6.1 | -41 | 8.2 | | | | | | | | |
| Jul-13 | -9.1 | -63 | 10.0 | -7.5 | -47 | 13.5 | | | | | | | | |
| Aug-13 | -4.3 | -38 | -3.6 | | | | | | | | | | | |
| Sep-13 | -9.6 | -62 | 15.0 | -8.8 | -57 | 13.0 | | | | | | | | |
| Oct-13 | -9.4 | -60 | 15.0 | -8.7 | -57 | 12.6 | | | | | | | | |
| Nov-13 | -9.8 | -56 | 22.2 | | | | | | | | | | | |
| Dec-13 | -20.7 | -156 | 9.9 | -21.2 | -152 | 16.9 | | | | | | | | |
| Jan-14 | -21.7 | -163 | 10.6 | -19.6 | -144 | 12.3 | | | | 4.8 | | | | 0.3 |
| Feb-14 | | | | | | | | | | 9.8 | | | | 0.7 |
| Mar-14 | -10.6 | -81 | 4.1 | | | | | | | 10.8 | | | | 0.8 |
| Apr-14 | -13.1 | -103 | 1.7 | -5.4 | -39 | 4.0 | | | | 17.0 | | | | 1.7 |
| May-14 | -12.2 | -81 | 16.1 | | | | | | | 32.1 | | | | 2.2 |
| Jun-14 | | | | | | | | | | 16.8 | | | | 1.0 |
| Jul-14 | | | | -6.1 | -38 | 11.2 | -9.4 | -65 | 10.5 | 21.9 | -9.0 | -64 | 8.6 | 1.5 |
| Aug-14 | | | | -5.1 | -35 | 5.7 | -9.3 | -65 | 9.3 | 11.1 | -9.0 | -64 | 7.9 | 0.3 |
| Sep-14 | | | | | | | -9.3 | -66 | 8.8 | 4.7 | -9.3 | -67 | 8.0 | 0.2 |
| Oct-14 | | | | -10.2 | -66 | 15.2 | -9.1 | -63 | 10.3 | 8.6 | -9.4 | -66 | 9.0 | 0.5 |
| Nov-14 | | | | -11.8 | -78 | 16.4 | -9.6 | -67 | 9.9 | 6.4 | -9.6 | -67 | 9.8 | 0.4 |
| Dec-14 | | | | -27.1 | -205 | 12.4 | -10.3 | -72 | 10.4 | 7.5 | -10.1 | -72 | 9.1 | 0.7 |
| Jan-15 | -14.6 | -109 | 7.8 | -12.6 | -99 | 2.2 | -10.0 | -70 | 10.5 | 9.9 | -9.9 | -69 | 10.4 | 0.8 |
| Feb-15 | -17.5 | -140 | 0.0 | -11.4 | -89 | 2.1 | -10.8 | -75 | 11.7 | 6.8 | -9.9 | -69 | 10.4 | 0.7 |
| Mar-15 | -15.3 | -113 | 9.4 | -11.3 | -83 | 7.9 | -10.6 | -73 | 12.3 | 18.6 | -10.3 | -71 | 11.5 | 1.5 |
| Apr-15 | -11.0 | -78 | 10.0 | -5.4 | -39 | 4.0 | -10.0 | -70 | 10.3 | 19.8 | -9.7 | -68 | 9.8 | 1.2 |
| May-15 | -8.5 | -56 | 12.0 | -9.6 | -65 | 11.9 | -9.3 | -66 | 8.3 | 9.6 | -10.7 | -75 | 11.2 | 0.4 |
| Jun-15 | -8.5 | -59 | 9.0 | -2.8 | -15 | 8.1 | -9.1 | -64 | 8.9 | 9.5 | -8.9 | -62 | 8.8 | 0.3 |
| Jul-15 | -10.0 | -69 | 11.0 | -5.8 | -34 | 12.3 | -9.1 | -65 | 8.2 | 5.2 | -8.8 | -63 | 6.7 | 0.1 |
| Aug-15 | -5.7 | -33 | 12.6 | -5.4 | -33 | 10.1 | -8.8 | -63 | 7.2 | 3.1 | -8.9 | -64 | 7.0 | 0.2 |
| Sep-15 | | | | -7.1 | -46 | 10.5 | -9.2 | -64 | 9.4 | 2.9 | -9.3 | -65 | 9.3 | 0.1 |
| Oct-15 | | | | -9.7 | -67 | 10.7 | -9.7 | -67 | 10.7 | 5.1 | -9.7 | -68 | 9.7 | 0.3 |
| Nov-15 | | | | -11.4 | -73 | 18.5 | -10.0 | -70 | 10.3 | 4.1 | -10.6 | -75 | 10.3 | 0.3 |
| Dec-15 | | | | -12.8 | -90 | 12.9 | -9.6 | -67 | 9.2 | 4.5 | -9.7 | -68 | 9.4 | 0.4 |
| Jan-16 | | | | -10.7 | -84 | 2.1 | -9.8 | -68 | 10.2 | 3.4 | -9.8 | -69 | 9.3 | 0.3 |
| Feb-16 | | | | | | | -10.1 | -70 | 10.5 | 6.8 | -9.7 | -69 | 9.1 | 0.4 |





| | | | | | | | | | | | | | |
|---|---|---|---|---|---|---|---|---|---|---|---|---|---|
| **Mar-16** | | | | -6.0 | -69 | | -10.0 | -70 | 10.1 | 5.5 | -9.5 | -68 | 8.5 | 0.4 |
| **Apr-16** | | | | -9.8 | -68 | 9.9 | -9.5 | -67 | 9.1 | 6.1 | -9.3 | -66 | 8.2 | 0.4 |
| **May-16** | | | | -15.4 | -114 | 9.8 | | | | 10.3 | -8.9 | -61 | 9.9 | 0.9 |
| **Jun-16** | -5.8 | -37 | 9.1 | 1.1 | -10 | | -9.2 | -63 | 10.2 | 48.1 | -9.0 | -62 | 9.9 | 4.4 |
| **Jul-16** | | | | | | | -9.0 | -63 | 9.1 | 8.9 | -8.2 | -61 | 5.1 | 0.5 |
| **Aug-16** | -11.0 | -70 | 18.0 | | | | -8.9 | -63 | 8.6 | 5.7 | -8.6 | -62 | 6.4 | 0.2 |
| **Sep-16** | -12.3 | -82 | 16.4 | -4.2 | -28 | 5.1 | -9.2 | -64 | 9.1 | 3.5 | -10.0 | -68 | 11.3 | 0.2 |
| **Oct-16** | -14.1 | -97 | 15.8 | -5.5 | -35 | 9.4 | -9.9 | -68 | 10.8 | 18.4 | -9.2 | -65 | 9.2 | 1.5 |
| **Nov-16** | -13.2 | -93 | 12.6 | -11.5 | -86 | 6.0 | -10.1 | -70 | 10.7 | 16.2 | -10.2 | -70 | 11.6 | 1.0 |
| **Dec-16** | -14.3 | -99 | 15.4 | -21.6 | -162 | 10.9 | -9.9 | -67 | 11.5 | 10.3 | -10.0 | -68 | 11.4 | 0.7 |





*Table 2*. δ-values in groundwater

| No. | Samples location | Type of sample | Litostructure | Mean altitude (m) | Mean (‰) | | d-excess |
|-----|------------------|----------------|---------------|-------------------|----------|----------|----------|
| | | | | | $\delta^{18}O$ | $\delta^2H$ | |
| 1 | Bănești | Well dug | Deluvial-proluvial deposits | 269 | -10.5 | -74 | 9.8 |
| 2 | Bilca | Well dug | Deluvial-proluvial deposits | 408 | -10.2 | -71 | 10.0 |
| 3 | Cacica | Well dug | Sand stone, marl, clay, gypsum | 444 | -9.5 | -67 | 9.3 |
| 4 | Cajvana | Well dug | Deluvial-proluvial deposits | 400 | -10.2 | -72 | 9.6 |
| 5 | Calafindești | Well dug | Marl clays with sand | 404 | -10.0 | -70 | 9.8 |
| 6 | Costâna | Well dug | Marl clays with sand | 305 | -10.2 | -72 | 9.8 |
| 7 | Cotu Dobei | Well dug | Deluvial-proluvial deposits | 265 | -10.6 | -74 | 10.8 |
| 8 | Dumbraveni | Well dug | Marl clays with sand | 303 | -9.9 | -71 | 8.2 |
| 9 | Ipotești | Well dug | Marl clays with sand | 328 | -10.0 | -71 | 9.0 |
| 10 | Itcani | Well dug | Gravels, sand | 280 | -9.4 | -66 | 9.1 |
| 11 | Izvoarele Suceavei | Well dug | Black Shale flish (Audia Nappe) | 940 | -10.4 | -72 | 10.5 |
| 12 | Moara | Well dug | Marl clays with sand | 377 | -9.8 | -68 | 10.6 |
| 13 | Părhăuți | Well dug | Deluvial-proluvial deposits | 303 | -10.4 | -73 | 10.4 |
| 14 | Pârteștii de Jos | Well dug | Gray marls and clay | 387 | -9.2 | -64 | 9.7 |
| 15 | Poieni Solca | Well dug | Marl clays with sand | 451 | -9.5 | -66 | 10.7 |
| 16 | Putna | Well dug | Shale flish | 574 | -9.7 | -68 | 10.0 |
| 17 | Rădăuți | Well dug | Deluvial-proluvial deposits | 372 | -9.8 | -68 | 10.2 |
| 18 | Siret | Well dug | Marl clays with sand | 303 | -9.5 | -67 | 9.2 |
| 19 | Soloneț | Well dug | Marl clays with sand | 356 | -9.6 | -67 | 9.4 |
| 20 | Văratec | Well dug | Marl clays with sand | 315 | -9.9 | -69 | 9.8 |
| 21 | Vicovu de Jos | Well dug | Gravels, sand | 430 | -9.8 | -68 | 10.8 |



***Table 3.*** Precipitation sources (ATL – Atlantic Ocean, MED – Mediterranean Sea, CONT-Continental Sources, BS – Black Sea, AR – Arctic Ocean, TP – Total Percentage of precipitation sources)

| Year | Season | d-excess | ATL | MED | CONT | BS | AR | TP |
|------|--------|----------|-----|-----|------|-----|-----|------|
|  | Winter | 13.0 | 0.0 | 18.4 | 24.7 | 9.6 | 0.0 | 52.7 |
|  | Spring | 12.4 | 19.5 | 8.8 | 48.8 | 0.0 | 0.0 | 77.1 |
| 2013 | Summer | 10.9 | 20.8 | 0.0 | 62.4 | 0.0 | 0.0 | 83.2 |
|  | Autumn | 12.8 | 9.4 | 0.0 | 45.9 | 0.0 | 28.4 | 83.6 |
|  | **Annual** | **12.3** | **12.4** | **6.8** | **45.5** | **2.4** | **7.1** | **74.2** |
|  | Winter | 14.6 | 19.5 | 0.0 | 23.9 | 0.0 | 0.0 | 43.5 |
|  | Spring | 4.0 | 30.1 | 12.1 | 33.7 | 0.0 | 9.4 | 85.2 |
| 2014 | Summer | 8.4 | 49.9 | 0.0 | 22.4 | 2.4 | 12.8 | 87.5 |
|  | Autumn | 15.8 | 4.2 | 0.0 | 33.7 | 0.0 | 40.3 | 78.2 |
|  | **Annual** | **10.7** | **25.9** | **3.0** | **28.4** | **0.6** | **15.6** | **73.6** |
|  | Winter | 5.6 | 22.3 | 24.0 | 20.1 | 0.0 | 0.0 | 66.4 |
|  | Spring | 7.9 | 34.4 | 0.0 | 40.6 | 5.1 | 0.0 | 80.2 |
| 2015 | Summer | 10.1 | 37.7 | 0.0 | 28.6 | 0.0 | 14.6 | 81.0 |
|  | Autumn | 13.2 | 15.9 | 14.8 | 47.5 | 0.0 | 0.0 | 78.2 |
|  | **Annual** | **9.2** | **27.6** | **9.7** | **34.2** | **1.3** | **3.7** | **76.4** |
|  | Winter | 7.5 | 30.9 | 4.1 | 9.7 | 0.0 | 0.0 | 44.8 |
|  | Spring | 9.9 | 22.8 | 0.0 | 47.7 | 11.2 | 0.0 | 81.7 |
| 2016 | Summer | - | 24.1 | 3.4 | 27.0 | 23.0 | 0.0 | 77.5 |
|  | Autumn | 6.8 | 12.7 | 0.0 | 59.3 | 0.0 | 13.2 | 85.1 |
|  | **Annual** | **8.1** | **22.6** | **1.9** | **35.9** | **8.6** | **3.3** | **72.3** |
|  | **Winter** | **10.2** | **18.2** | **11.6** | **19.6** | **2.4** | **0.0** | **51.8** |
| Average | **Spring** | **8.6** | **26.7** | **5.2** | **42.7** | **4.1** | **2.3** | **81.1** |
| 2013-2016 | **Summer** | **9.8** | **33.1** | **0.9** | **35.1** | **6.4** | **6.9** | **82.3** |
|  | **Autumn** | **12.2** | **10.5** | **3.7** | **46.6** | **0.0** | **20.5** | **81.3** |



**Figure 1.** Location of the study site in Europe (a) and Romania (b). The map in (b) shows the annual mean temperature trend over Romania for the period 1961 – 2013. Panel (c) shows the location of the studied drainage basin with the sampling points locations.





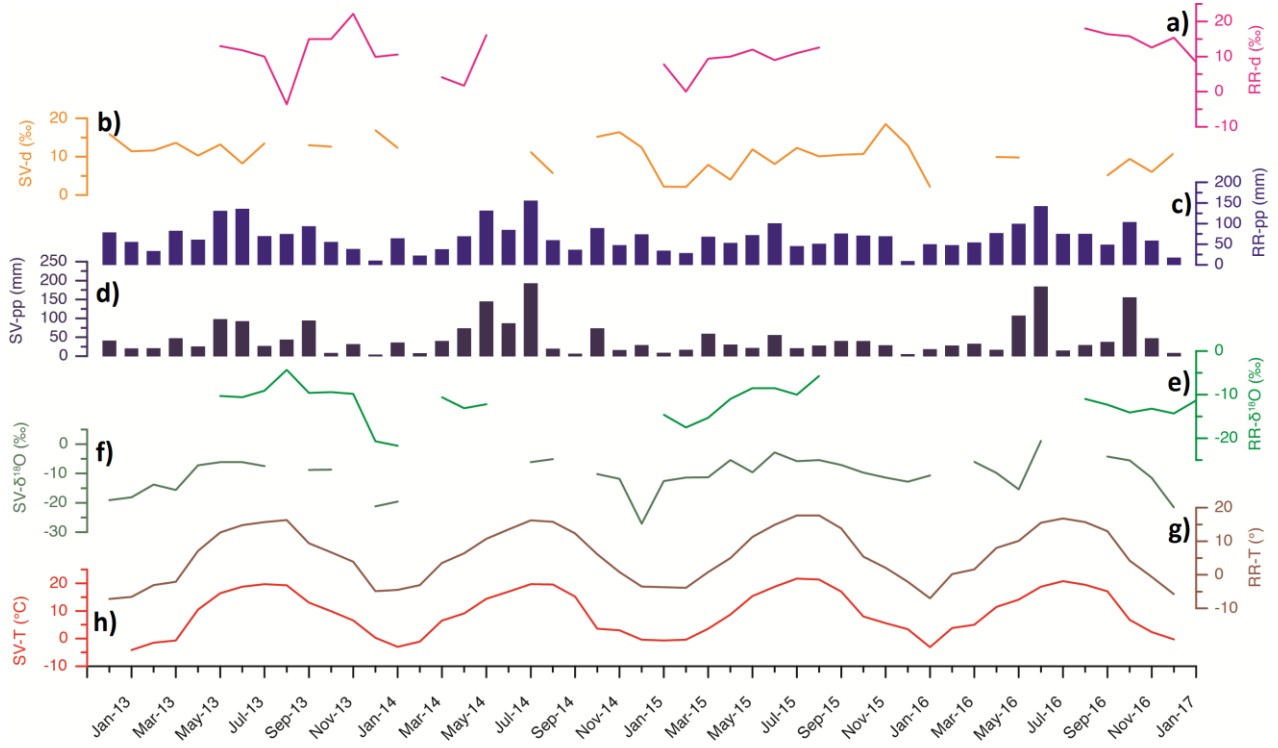

**Figure 2:** Monthly time series d-excess (d) at a) Rarău station (RR) and b) Suceava station (SV), precipitation amount (pp) at RR (c) and SV (d), precipitation ($\delta^{18}O$) at RR (e) and SV (f) and air temperature (T) at RR (g) and SV (h).



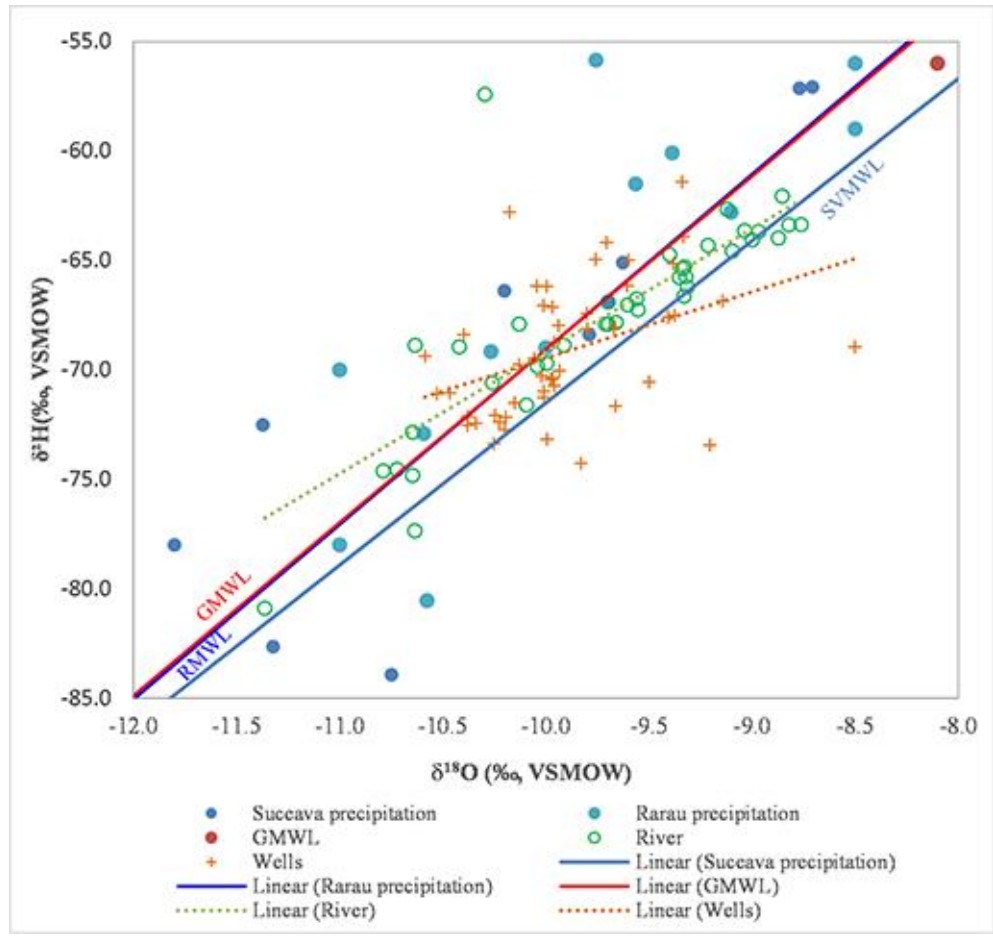

**Figure 3.** LMWLs and RWL for precipitation (blue line), surface (green line) and groundwaters (orange line) in the Suceava Watershed.



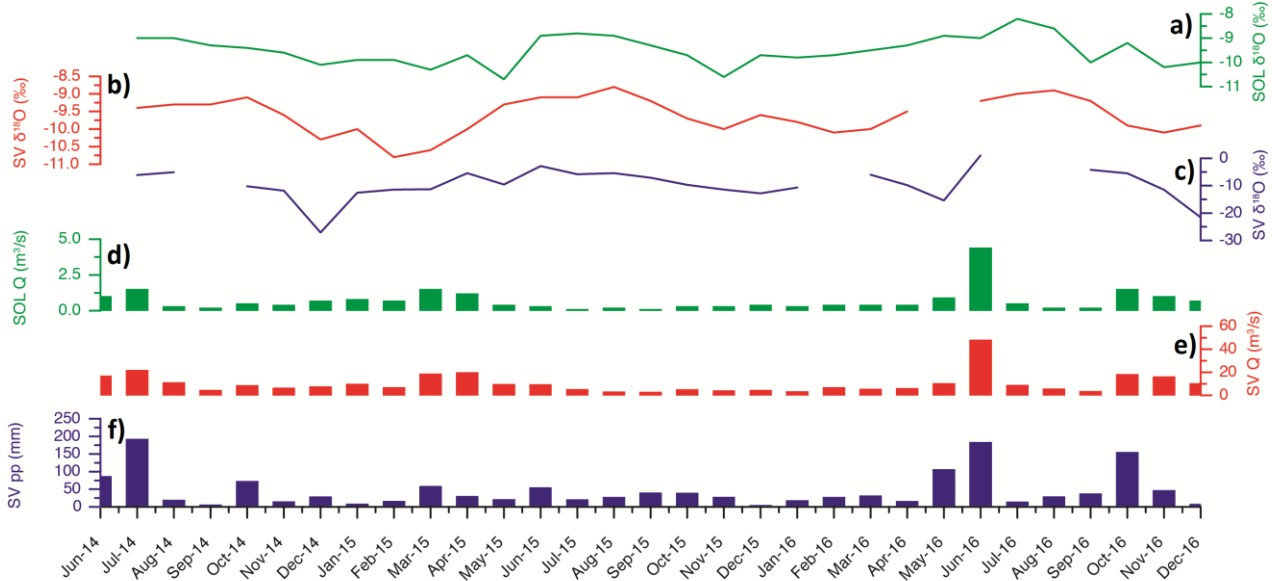

**Figure 4.** Monthly time series of river δ¹⁸O of a) Soloneț River (SOL) and b) Suceava River (SV) δ¹⁸O in precipitation at Suceva station (c), river discharge (Q) of SOL (d) and SV (e) rivers and precipitation amount (pp) at Suceava station (f).




**Figure 5.** (a) The *high* (TT >1 standard deviation) composite map between winter mean temperature (TT) at Suceava station and winter (DJF) geopotential height at 500 mb (Z500 – shaded areas) and winter 500 mb wind vectors (arrows); (b) *low* (TT >0.75 standard deviation) composite map between winter mean temperature (TT) at Suceava station and winter geopotential height at 500 mb (Z500 – shaded areas) and winter 500 mb wind vectors (arrows); (c) as in (a) but for summer TT and summer Z500 and (d) as in (b), but for summer TT and summer Z500.



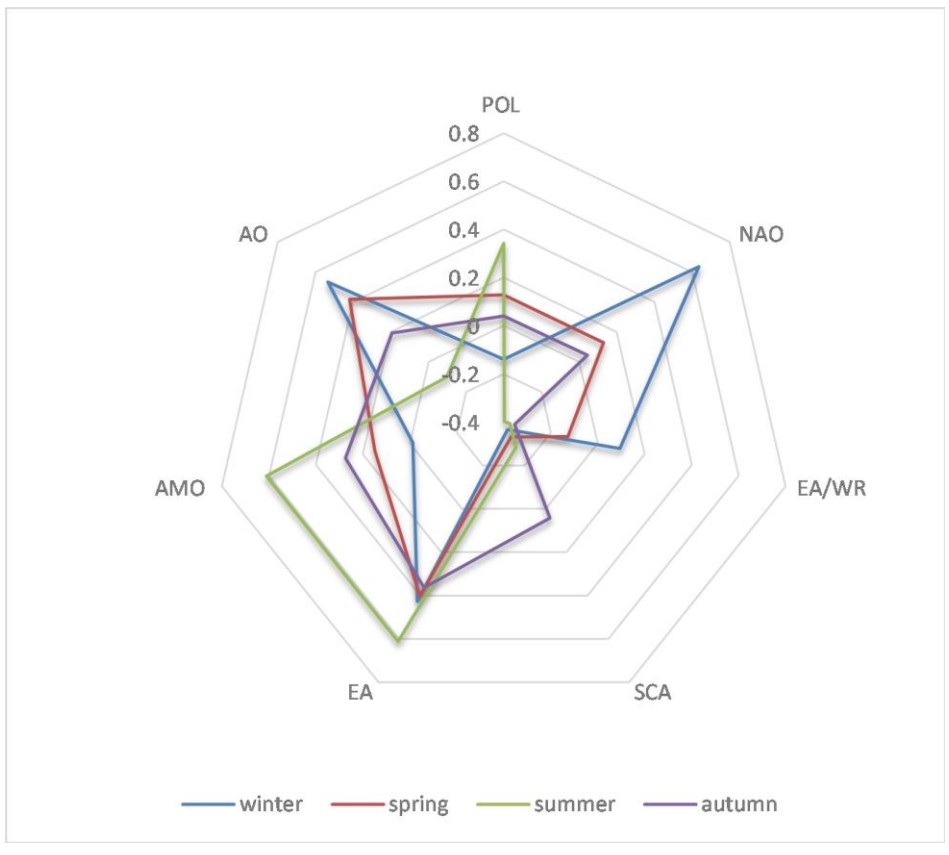

**Figure 6.** Correlation coefficient between air temperature at Suceava station and teleconnection patterns for winter (blue line), spring (red line), summer (green line) and autumn (purple line) seasons.





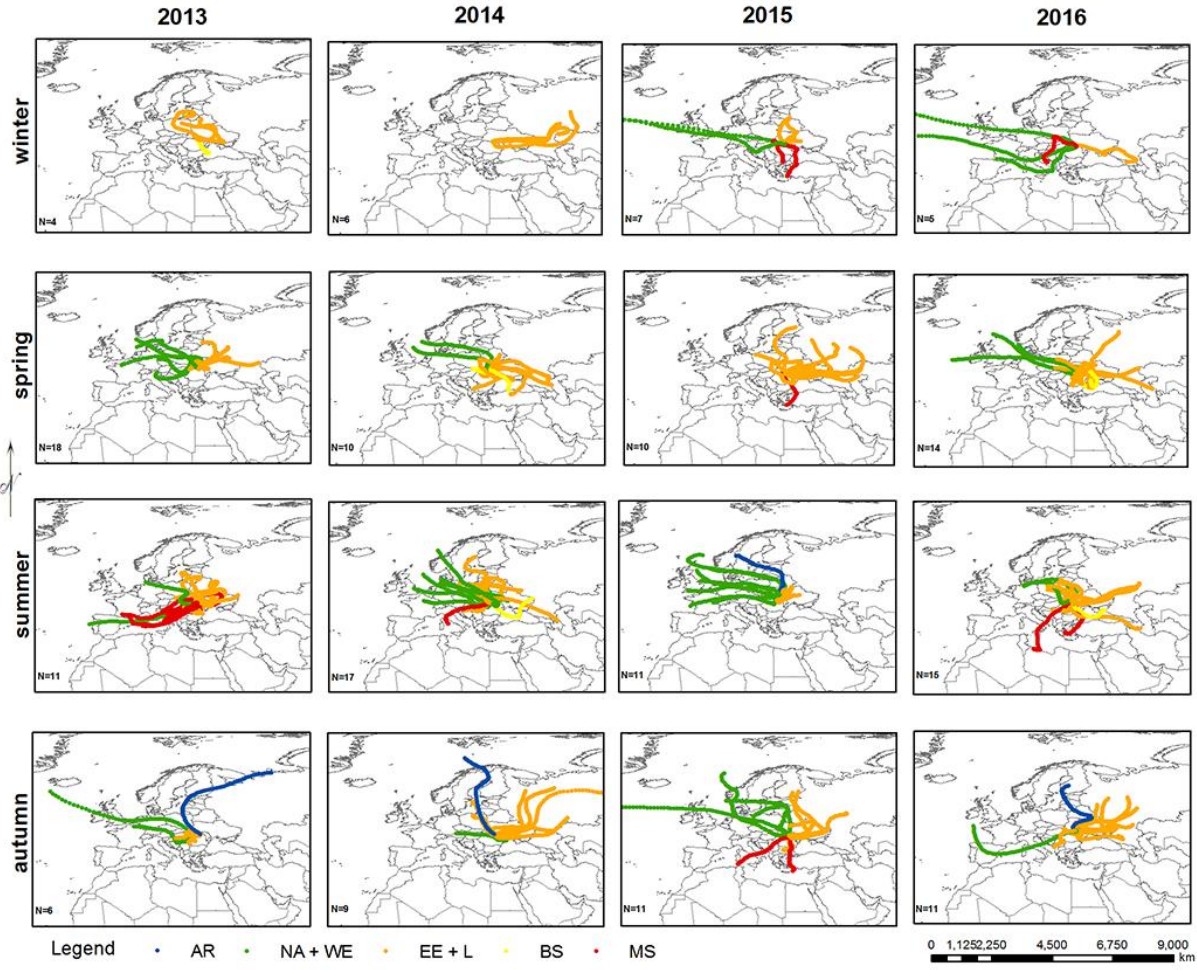

**Figure 7**. Modeled trajectories for single precipitation events (2013 – first panel, 2014 – second panel, 2015 –third panel and 2016 – last panel) at Suceava station based on the HYSPLIT model.



**Figure 8.** Seasonal distribution of δ¹⁸O (left panel) and d-excess (right panel) in groundwater in the Suceava River watershed.