# Peer review of "Stable isotope investigation of groundwater recharge in the Carpathian Mountains, East-Central Europe"

_Hydrology and Earth System Sciences, 2018_

## Referee Comment (RC1) · Anonymous Referee #1 · 15 Feb 2018

Unfortunately, I cannot recommend that this paper be published in its present form. The paper's title and introduction suggest that it is a recharge study; however, that is probably the least discussed of the material in the paper.

The only text that pertains to recharge is at the end of section 5.2 (page 11, lines 11-23) where the main conclusions appear to be that there are few discernible patterns in the stable isotopes and this implies that the groundwater is recharged locally and may be around six months old. The first point is self-evident – recharge is always a local process, what the authors may mean is that there has been little mixing in the aquifers subsequent to recharge (although there is little data presented to test that).

4

[Figure]

The assumptions behind the approximate residence time are also not justified and it is speculative.

The text in this section (page 11, lines 19-23) are also contradictory. To estimate the residence times or to understand the sources of water using geochemistry, there needs to be little or no mixing in the aquifers. The statement that there is "slow underground flow on the same direction of the aquifer" seems to imply that local recharge is mixed with laterally-flowing groundwater, which would not allow the geochemistry to be used in this way.

There is also insufficient detail on the hydrogeology. Specifically,

- the flow systems are not defined

- there is no indication of where the recharge areas are (Fig. 1 show many of the wells to be located in valleys, are these really in discharge areas?)

- there is no indication of wells depths, water table levels etc. If one wishes to sample recharge in wells, they need to be screened at the water table.

- the only mention of methodology relating to the groundwater is actually "The spatial variability of stable isotopes in groundwater was spatially mapped using the ordinary kriging method (Nas and Berktay, 2010)."

- 157 groundwater samples are mentioned in the abstract, but Table 2 list only 21 (and a similar number are on Fig. 3). If you are going to claim a large data set you need to present it and use it.

The paper is mainly concerned with using stable isotopes to understand moisture sources. This is not really my field of expertise although I can follow the logic of the discussion. The authors would be better to cast the paper in that way with perhaps a sentence or two at the end stating that this is important information for understanding recharge or residence times in rivers etc.

[Figure]

The paper is also very specific and there are no indications of how it relates to previous work on this topic in general or the region (there are papers by Bojar et al, 2017 and Bottyan et al., 2017 that look to cover similar topics). The conclusions also do not convey any general importance and it is not clear what a reader working on these topics elsewhere would get from the paper.

At the very least, the authors need to rewrite the paper so that the title and introduction more clearly reflect the contents. Even then, the paper appears to have limited scope and may be more suited to a regional journal.

––––––––––––––––––––––––––––––––––––––

---

## Referee Comment (RC2) · Anonymous Referee #2 · 9 Mar 2018

General commments The present manuscript is bringing new data for a major catchment area of the northern Moldavia, Romania. Th e data includes: local meteoric water lines, well distribution and isotopic compositions of goundwaters, seasonal river isotopic compositions, seasonal variations in humidity and temperature, amount of precipitation as well as circulation patterns. Such investigations are largely missing for the region and is important for the region to see such an impressive compilation of data and seasonal temperature distribution and moisture circulation models.

Specific comments Before publication, specific issues should be improved:

As the manuscript address regional circulation patterns, this should be also reflected

in the title.

For Table 1 the abbreviations used for columns should be explained in the caption of the Table.

Mention if the calculated mean yearly isotopic compositions are amount weighted or not.

For the Figure 3 Legend and Plot: In the Legend, below the figure, left column, there is a red filled point explained as representing GMWL. The red point is not GMWL, please explain the meaning of the red point correctly. Avoid using each time "linear" for explanations in the Legend. There are two abbreviations within the plot, SVMWL and RMWL, but just one blue line is displayed. Also these two abbreviations are not mentioned in the Legend. For river waters, I admit that there is mainly a linear regression trend. For well waters there is not a single regression, the pattern is more complicated, probably you was sampling several aquifers situated at various depths. This should be insert in the discussions as well. The blue line indicating local meteoric water line is not reflecting the regression for the blue filled points (local precipitations), check data. After checking once more the position of the local meteoric water line (LMWL), discuss the data plotting left of the local meteoric water line. Which should be the reason(s) for this?

In the Introduction you mention that "The links between the stable isotope distribution of O and H in water and climate are well understood in general principles and these can be locally applied to distinguish between different moisture sources and tracks, seasonal contribution to river and groundwater recharge (ref), post-precipitation processes (e.g., evaporation) etc. However, so far, no such studies have been performed in our study area, and, as a matter of fact, in Romania, except for a few studies aimed at understanding the stable isotope composition of precipitation in Western Romania (Bojar et al., 2009; Bojar et la. 2017; DrăguÈŹin et al., 2017)." This is not correct; please look once more at the papers of Bojar et al., 2017 and DrăguÈŹin et al., 2017.

Both papers are investigating and discussing the relationship between precipitations and groundwaters for clastic and karstic aquifers, respectively. The investigated area in your manuscript is situated like 600 km away from those areas and according to your data show a different moisture circulation pattern. Please remodel the paragraph in the light of these facts.

You have the data necessary in order to insert in the text, for precipitation, the Dansgaard equations between temperature and isotopic compositions. In the reference list Dansgaard paper is included but a reference to that paper is missing from the manuscript text.

The statement in the Conclusion "the main precipitation sources are located eastwards from the sampling site (in the East-European Plain and the Black Sea)" is not supported by the data shown in Table 3. Also the role of local recycling is missing, I suggest Table 3 should be interpreted in a more moderate style.

In Table 3 caption, please include a short statement about the method you used in the calculation of the precipitation source percentages. Also add an explicit paragraph in the Methods about this topic.

---

## Referee Comment (RC3) · Anonymous Referee #3 · 29 Mar 2018

It is a pity, but there are basic problems with this manuscript, therefore I can't recommend its publication in this state.

1. According to the title the main focus of the manuscript (MS) is the study of the groundwater recharge. The authors took samples from dug wells for characterizing the shallowest groundwater, but the sampling method they applied was not appropriate. They simple took grab samples 30-50 cm below the surface of the water in the dug wells. Water in a dug well is in direct contact with the atmosphere, and so it may evaporate easily, which modify both chemistry and isotopic characteristics. Looking at the groundwater data on Figure 3 we can easily recognize that several water samples suf-

fered evaporation effect (they are far below the Local Meteoric Water Lines). Actually the slope of the trend line of groundwater samples has got the lowest value, which is another indication for evaporation effect. This entire means that the collected groundwater samples are not representative of the shallowest groundwater. The proper way of taking representative sample of groundwater from dug well involves the removal all the water from the well and the newly infiltrated water can be used for sampling.

2. If we want to determine whether the shallowest groundwater is locally infiltrated or it was infiltrated at a higher elevation area, minimum we need a conceptual groundwater flow model. This is completely missing from the manuscript. Having been identified the local recharge areas we can characterize the isotopic composition of the locally infiltrated water. On the local, intermediate or regional discharge areas the locally infiltrated water necessarily mixes with the discharging groundwater. Once we know the characteristics of the locally infiltrated water, we can study this mixing process.

3. Major part of the manuscript deals with precipitation including its isotopic characteristics. But: stable isotope time series are discontinuous for both stations, Rarău and Suceava, see Table 1. In case of Rarău there are long periods with no data, e.g. from December 2012 to April 2013, or from June 2014 to December 2014, or from September 2015 to May 2016. The situation for the Suceava station is far better, but there are several months (actually 10) without any data. I hardly believe that there was no precipitation for so long periods of time (the MS doesn't mention any reason for lack of data). This entire means that the precipitation isotope data theoretically don't describe well the local precipitation. This data set can be used for calculating the first approximation of the LMWLs, but inadequate for calculating the multiannual means of delta values.

4. Not having representative groundwater samples, neither proper mean delta values of local precipitation the "Stable isotope investigation of groundwater recharge" is hopeless, or at least inappropriate.

5. HYSPLIT: I am not experienced in this field, so I have discussed this part with two of my colleagues, who are applying this method in their research work. They have confirmed my feeling that modelling at only one level (500mb) is not enough. Modelling at three levels is the most common situation in these kinds of publications (recently). Furthermore, the specific humidity along the trajectory was not determined, so the source region of the air mass was determined, but not the source region of the vapor!

---

## Author Comment (AC1) · 25 Apr 2018

We would like thank the referee for the fast response and to the recommendations and we are grateful for the comments on how it can be further improved. We provide below a general response to the reviewer's comments/suggestions.

Comment: The paper's title and introduction suggest that it is a recharge study; however, that is probably the least discussed of the material in the paper. The only text that pertains to recharge is at the end of section 5.2 (page 11, lines 11-23) where the main conclusions appear to be that there are few discernible patterns in the stable isotopes and this implies that the groundwater is recharged locally and may be around six

months old. The first point is self-evident – recharge is always a local process, what the authors may mean is that there has been little mixing in the aquifers subsequent to recharge (although there is little data presented to test that). The assumptions behind the approximate residence time are also not justified and it is speculative. The text in this section (page 11, lines 19-23) are also contradictory. To estimate the residence times or to understand the sources of water using geochemistry, there needs to be little or no mixing in the aquifers. The statement that there is "slow underground flow on the same direction of the aquifer" seems to imply that local recharge is mixed with laterally-flowing groundwater, which would not allow the geochemistry to be used in this way. There is also insufficient detail on the hydrogeology. Specifically, - the flow systems are not defined - there is no indication of where the recharge areas are (Fig. 1 show many of the wells to be located in valleys, are these really in discharge areas?) - there is no indication of wells depths, water table levels etc. If one wishes to sample recharge in wells, they need to be screened at the water table. - the only mention of methodology relating to the groundwater is actually "The spatial variability of stable isotopes in groundwater was spatially mapped using the ordinary kriging method (Nas and Berktay, 2010)." - 157 groundwater samples are mentioned in the abstract, but Table 2 list only 21 (and a similar number are on Fig. 3). If you are going to claim a large data set you need to present it and use it. The paper is mainly concerned with using stable isotopes to understand moisture sources. This is not really my field of expertise although I can follow the logic of the discussion. The authors would be better to cast the paper in that way with perhaps a sentence or two at the end stating that this is important information for understanding recharge or residence times in rivers etc. The paper is also very specific and there are no indications of how it relates to previous work on this topic in general or the region (there are papers by Bojar et al, 2017 and Bottyan et al., 2017 that look to cover similar topics). The conclusions also do not convey any general importance and it is not clear what a reader working on these topics elsewhere would get from the paper.

Response: We fully agree that the title and the general organization of the text do

not clearly lead to the idea enounced in the title. We attempted a recharge study, the hypothesis being that we would be able to 1) disentangle between various moisture sources and tracks feeding precipitation in the study area and analyze their controlling factors, 2) track how precipitation water feeds local aquifers and 3) analyze the relationship between precipitation, river and ground waters; the overall aim being to provide policymakers a first tool to be used (and improved) to asses groundwater resources. There is a general lack of understanding in the region on how science can be put behind decisions – and we made an attempt to fill it. Nevertheless, the comments we have received have offered us a better "redline" to be followed towards achieving this goal and below we show how this will be used to improve the study and the paper. Thus, our article is now recast to address the points above. As such, we start with an improved description of the processes controlling the stable isotope composition of precipitation and continue with a thoroughly revised section on groundwater stable isotope geochemistry. This includes a better description of the local flow systems (with delineation of the recharge areas, description of the wells – depth, water table level and fluctuations, type of water usage removal – bucket vs. pump etc). Further, the text is directed towards linking the stable isotope composition of groundwater with that of precipitation and a discussion of the processes behind the variability in stable isotope composition in groundwater in relation with precipitation and rivers. We have separated the wells into two groups – located near the rivers and farther away - and discuss to what extent the exchange with river waters is seen in the stable isotope composition if groundwater and river water. We have attempted - and subsequently discarded – the calculation of isotope enabled mass balance as the data sets on discharge was very limited. As such, the paper now has a slightly different focus that makes a better use of the dataset we have, strengthening the conclusion, removing less-well supported claims and providing an assessment tool that could be used (and improved) for regions where similarly limited data sets are (or could be) obtained.

---

## Author Comment (AC2) · 25 Apr 2018

We would like thank the referee for the fast response and to the recommendations and we are grateful for the comments on how it can be further improved. We provide below a point by point response to the reviewer's comments/suggestions.

Comment: As the manuscript address regional circulation patterns, this should be also reflected in the title. Response: Both the title and the keywords will be modified to reflect the (improved) new structure of the text.

Comment: For Table 1 the abbreviations used for columns should be explained in the

caption of the Table. Response: For Table 1 the abbreviations used for columns will be changed to more descriptive ones (O = $\delta18$O, H = $\delta2$H) or explained in the caption (d = d-excess, Q = liquid flow, RR = Rarău station, SV = Suceava station, SVR = Suceava River, SOL =SoloneÈŻ River).

Comment: Mention if the calculated mean yearly isotopic compositions are amount weighted or not. Response: No, they are not. We generally favor the usage of raw data as much as possible. Further, we are also providing the precipitation amount, and as such, readers interested can calculate their own amount-weighted values.

Comment: For the Figure 3 Legend and Plot: In the Legend, below the figure, left column, there is a red filled point explained as representing GMWL. The red point is not GMWL, please explain the meaning of the red point correctly. Avoid using each time "linear" for explanations in the Legend. There are two abbreviations within the plot, SVMWL and RMWL, but just one blue line is displayed. Also these two abbreviations are not mentioned in the Legend. For river waters, I admit that there is mainly a linear regression trend. For well waters there is not a single regression, the pattern is more complicated, probably you was sampling several aquifers situated at various depths. This should be insert in the discussions as well. The blue line indicating local meteoric water line is not reflecting the regression for the blue filled points (local precipitations), check data. After checking once more the position of the local meteoric water line (LMWL), discuss the data plotting left of the local meteoric water line. Which should be the reason(s) for this? Response: this is just an artifact of drawing the GMWl. We have plotted a couple of points along a line defined by the GMWL's equation (ïĄď2H = ïĄď18O+10) and then plotted the liner fit to these points. This linear fit is the GMWL Figure 3 is a zoomed-in version of the original, larger, figure, and as such, one of the points is shown. We will delete it. Further, in the final manuscript we will avoid using each time "linear" for explanations in the Legend, this will be replaced with abbreviations, which will be explained in the caption of Figure 3. RMWL (Rarau Meteoric Water Line) represent the dark blue line but it almost perfectly overlaps the red line (GMWL),

while SVMWL (Suceava Meteoric Water Line) is the light blue line. We wil use colors to make all lines visible. Indeed, the groundwater was sampled at various depths and from aquifers located in different lithologies – as such, drawing a linear fit does bring confusion, rather than clarification. We will remove it. The points located above (left) the LMWL reflect samples evaporated in summer (evaporation of falling raindrops in dry atmosphere).

Comment: In the introduction you mention "The links between the stable isotope distribution of O and H in water and climate are well understood in general principles and these can be locally applied to distinguish between different moisture sources and tracks, seasonal contribution to river and groundwater recharge (ref), post-precipitation processes (e.g., evaporation) etc. However, so far, no such studies have been performed in our study area, and, as a matter of fact, in Romania, except for a few studies aimed at understanding the stable isotope composition of precipitation in Western Romania(Bojar et al., 2009; Bojar et la. 2017; DrăguÈŹin et al., 2017)." This is not correct; please look once more at the papers of Bojar et al., 2017 and DrăguÈŹin et al., 2017. Both papers are investigating and discussing the relationship between precipitations and groundwaters for clastic and karstic aquifers, respectively. The investigated area in your manuscript is situated like 600 km away from those areas and according to your data show a different moisture circulation pattern. Please remodel the paragraph in thelight of these facts Response: We will remodel the paragraph to balance the contribution of the studies by Bojar et al. (2017) and DrăguÈŹin et al. (2017) – the later of which was also made by our research group.

Comment: You have the data necessary in order to insert in the text, for precipitation, the Dansgaard equations between temperature and isotopic compositions. In the reference list Dansgaard paper is included but a reference to that paper is missing from the manuscript text Response: We will include Dansgaard in text in the Introduction section (page 4, line 2), where we have left only "ref" in the main text.

Comment: The statement in the Conclusion "the main precipitation sources are located

eastwards from the sampling site (in the East-European Plain and the Black Sea)" is not supported by the data shown in Table 3. Also the role of local recycling is missing, I suggest Table 3 should be interpreted in a more moderate style. Response: In table 3 we have combined locally recycled moisture with the Easterly (which we named "continental") sources. We have separated these in table 3 now and have rephrased the text accordingly. Our data now shows that easterly sources (see the orange trajectories in Fig. 7) account for ca. 20 % of moisture, on par with Atlantic ones, with Mediterranean ones coming third in importance. The Black Sea and locally recycled moisture are coming fourth in importance, but with a larger variability from year to year. The text will include all this information.

Comment: In Table 3 caption, please include a short statement about the method you used in the calculation of the precipitation source percentages. Also add an explicit paragraph in the Methods about this topic. Response: For calculation of the precipitation source percentages we have calculate the sum of amount of precipitation from one direction for each analyzed month. For this we used the next formula: PSP = (*100)/Pmonth where, PSP - precipitation source percentages PS – amount of precipitation in one direction Pmonth - monthly total rainfall in a month.

---

## Author Comment (AC3) · 25 Apr 2018

We would like thank the referee for the fast response and to the recommendations and we are grateful for the comments on how it can be further improved. We provide below a point by point response to the reviewer's comments/suggestions.

Comment 1: According to the title the main focus of the manuscript (MS) is the study of the groundwater recharge. The authors took samples from dug wells for characterizing the shallowest groundwater, but the sampling method they applied was not appropriate. They simple took grab samples 30-50 cm below the surface of the water in the dug wells. Water in a dug well is in direct contact with the atmosphere, and so it may

evaporate easily, which modify both chemistry and isotopic characteristics. Looking at the groundwater data on Figure 3 we can easily recognize that several water samples sufered evaporation effect (they are far below the Local Meteoric Water Lines). Actually the slope of the trend line of groundwater samples has got the lowest value, which is another indication for evaporation effect. This entire means that the collected ground-water samples are not representative of the shallowest groundwater. The proper way of taking representative sample of groundwater from dug well involves the removal all the water from the well and the newly infiltrated water can be used for sampling.

Response 1: We have used the standard strategy of collecting water from dug wells (Hunt et al. 2005). Further, we have removed the GMWL given the very similar stable isotope values in the samples, which translate into a line that is unrepresentative. While the "cloud" of groundwater data might suggest evaporated samples, this is not the case here. The d-excess values of the groundwater samples is usually between 9 and 10, close to the average of precipitation, with only 5 out of 88 being lower than 9 (and 3 from the same well), thus indicating extremely limited (or absent) evaporation.

Comment 2: If we want to determine whether the shallowest groundwater is locally infiltrated or it was infiltrated at a higher elevation area, minimum we need a conceptual groundwater flow model. This is completely missing from the manuscript. Having been identified the local recharge areas we can characterize the isotopic composition of the locally infiltrated water. On the local, intermediate or regional discharge areas the locally infiltrated water necessarily mixes with the discharging groundwater. Once we know the characteristics of the locally infiltrated water, we can study this mixing process.

Response 2: We have switched both the approach and the general organization of the text. . We fully agree that the title and the general organization of the text do not clearly lead to the idea enounced in the title. We attempted a recharge study, the hypothesis being that we would be able to 1) disentangle between various moisture sources and tracks feeding precipitation in the study area and analyze their controlling

factors, 2) track how precipitation water feeds local aquifers and 3) analyze the relation-ship between precipitation, river and ground waters; the overall aim being to provide policymakers a first tool to be used (and improved) to asses groundwater resources.

Comment 3: Major part of the manuscript deals with precipitation including its isotopic character- istics. But: stable isotope time series are discontinuous for both stations, Rarău and Suceava, see Table 1. In case of Rarău there are long periods with no data, e.g. from December 2012 to April 2013, or from June 2014 to December 2014, or from September 2015 to May 2016. The situation for the Suceava station is far better, but there are several months (actually 10) without any data. I hardly believe that there was no precipitation for so long periods of time (the MS doesn't mention any reason for lack of data). This entire means that the precipitation isotope data theoretically don't describe well the local precipitation. This data set can be used for calculating the first approx- imation of the LMWLs, but inadequate for calculating the multiannual means of delta values.

Response 3: The lack of samples from the Suceava station is due to the extremely low amounts of rainfall in the relevant months, while at the Rarău station the lack of most samples is due to possible evaporation of the samples after collection (they were stored at room temperature for more than two weeks and we have decided not to include them in the study).

Comment 4: Not having representative groundwater samples, neither proper mean delta values of local precipitation the "Stable isotope investigation of groundwater recharge" is hopeless, or at least inappropriate.

Response 4: We agree on the inappropriateness of the title. As discussed above, we have changed the general approach and structure of the text, and hence the title was changed to - Moisture sources, transport and hydrological processes in Carpathian Mountains, Central East Europe.

Comment 5: HYSPLIT: I am not experienced in this field, so I have discussed this part
with two of my colleagues, who are applying this method in their research work. They have confirmed my feeling that modelling at only one level (500mb) is not enough. Modelling at three levels is the most common situation in these kinds of publications (recently). Furthermore, the specific humidity along the trajectory was not determined, so the source region of the air mass was determined, but not the source region of the vapor!

Response 5: We have modeled the HYSPLIT-based trajectories at three levels (500, 1000 and 1500 m AGL), however, the differences were negligible, with little to know impact on the results. We have plotted the trajectories at 1000 m AGL and used these for calculating the sources. Now, the sources. The model was run for days with significant precipitation (>3 mm) in order to minimize the effect of sub-cloud evaporation of raindrops on the stable isotope composition of meteoric water. For each trajectory, we have acquired hourly atmospheric pressure, potential and environmental temperature, precipitation and relative humidity. Along every trajectory we have calculated (using HYSPLIT output data) the specific humidity using standard equations (Baldini et al., 2010, Krklec et al., 2014, Sodemann et al., 2008). Further, we have derived the moisture uptake using the specific humidity following Sodemann et al. (2008) and Krkelec et la (2014). Interestingly, a good correlation was found between the source region of the air masses and the source region of the vapor for moisture derived from both the three "wett" sources (Atlantic Ocean, Mediterranean and Black Sea) and the dry continental one. Further, in the case of local trajectories, during summer, locally evaporated water was more important in terms of amount and stable isotope composition than that from the source region of the air masses. However, in winter, the stable isotope composition of the moisture was more conservative, "preserving" the original signal.